# Modular and stereoselective synthesis of tetrasubstituted vinyl sulfides leading to a library of AIEgens

Xun-Shen Liu [1,2], Zhiqiong Tang[1], Zhiming Li [3✉], Mingjia Li[1], Lin Xu[1] & Lu Liu [1,2✉]

Tetraarylethylenes exhibit intriguing photophysical properties and sulfur atom frequently play a vital role in organic photoelectric materials and biologically active compounds. Tetra-substituted vinyl sulfides, which include both sulfur atom and tetrasubstituted alkenes motifs, might be a suitable skeleton for the discovery of the new material molecules and drug with unique functions and properties. However, how to modular synthesis these kinds of compounds is still challenging. Herein, a chemo- and stereo-selective Rh(II)-catalyzed [1,4]-acyl rearrangements of α-diazo carbonyl compounds and thioesters has been developed, providing a modular strategy to a library of 63 tetrasubstituted vinyl sulfides. In this transformation, the yield is up to 95% and the turnover number is up to 3650. The mechanism of this reaction is investigated by combining experiments and density functional theory calculation. Moreover, the "aggregation-induced emission" effect of tetrasubstituted vinyl sulfides were also investigated, which might useful in functional material, biological imaging and chemi-calnsing via structural modification.

[1] School of Chemistry and Molecular Engineering, East China Normal University, 500 Dongchuan Road, Shanghai 200241, China. [2] Shanghai Engineering Research Center of Molecular Therapeutics and New Drug Development, East China Normal University, 3663 N Zhongshan Road, Shanghai 200062, China. [3] Department of Chemistry, Fudan University, 2005 Songhu Road, Shanghai 200438, China. ✉email: zmli@fudan.edu.cn; lliu@chem.ecnu.edu.cn

Organosulfur chemistry has attracted more and more attention and brought increasing significance because carbon–sulfur bonds widely exist in nature, including biologically active molecules[1–4], pharmaceuticals[5], agrochemicals[6], materials[7], flavors, and fragrances[8], and even food ingredients[9]. Sulfur-containing motifs frequently play a vital role in organic photoelectric materials and biologically active compounds because S atom possesses strong polarizability and hyperpolarizability. Meanwhile, since the concept of "aggregation-induced emission" (AIE) was disclosed by Tang and co-workers[10–21] in 2001, the development of new AIE luminogens (AIEgens) has attracted significant attention during the past few years because of the enormous application potential in organic light-emitting diodes (OLEDs)[22,23], biological imaging[24,25], chemical sensors[26,27], cancer ablation[28,29], and theranostics[30,31]. Thus, the introducing of S atom into tetrasubstituted alkenes, one type of the classical AIE gens, might be an efficient way for the development of new material molecules with unique functions and properties (Fig. 1a).

Diazo compounds, especially α-diazocarbonyl compounds which are easily prepared and handled, are highly important reagents in synthetic chemistry, because they have high and versatile reactivity which have been used in a series of carbene transfer reactions including X-H (X = O, N, S, C, etc.) insertion, cyclopropanation, and ylide formation under the catalysis of transition metals[32–37]. They are also used frequently for the synthesis of sulfides via the construction of C–S bond[38–42]. In this regard, rearrangements via sulfonium ylides offer a straightforward and versatile way of accessing sulfide with carbonyl groups, which include [1,2]-rearrangement (Stevens rearrangement[43–46]) and [2,3]-rearrangement (Doyle-Kirmse rearrangement[47–52], and Sommelet-Hauser rearrangement[53–55]). However, these reactions could only afford the products with C(sp³)–S bond (Fig. 1b). Although Wang et al. reported an aromatic indolyl C(sp²)–S bond formation from the reaction of 3-diazoindol-2-imines with thioesters[56], the formation of vinyl C(sp²)–S bond via the sulfonium ylides rearrangement of the reaction of sulfur-containing compounds with diazo compounds is still unknown.

In this work, based on our continuous interest in transition-metal catalyzed carbene transfer reaction of α-diazo carbonyl compounds[57–59], we describe herein that structurally diverse tetrasubstituted vinyl sulfides (TVSs) are modularly assembled for the first time by a rhodium-catalyzed ylide formation/[1,4]-acyl transfer reaction of S-acyl thiol and diazo compounds (Fig. 1c).

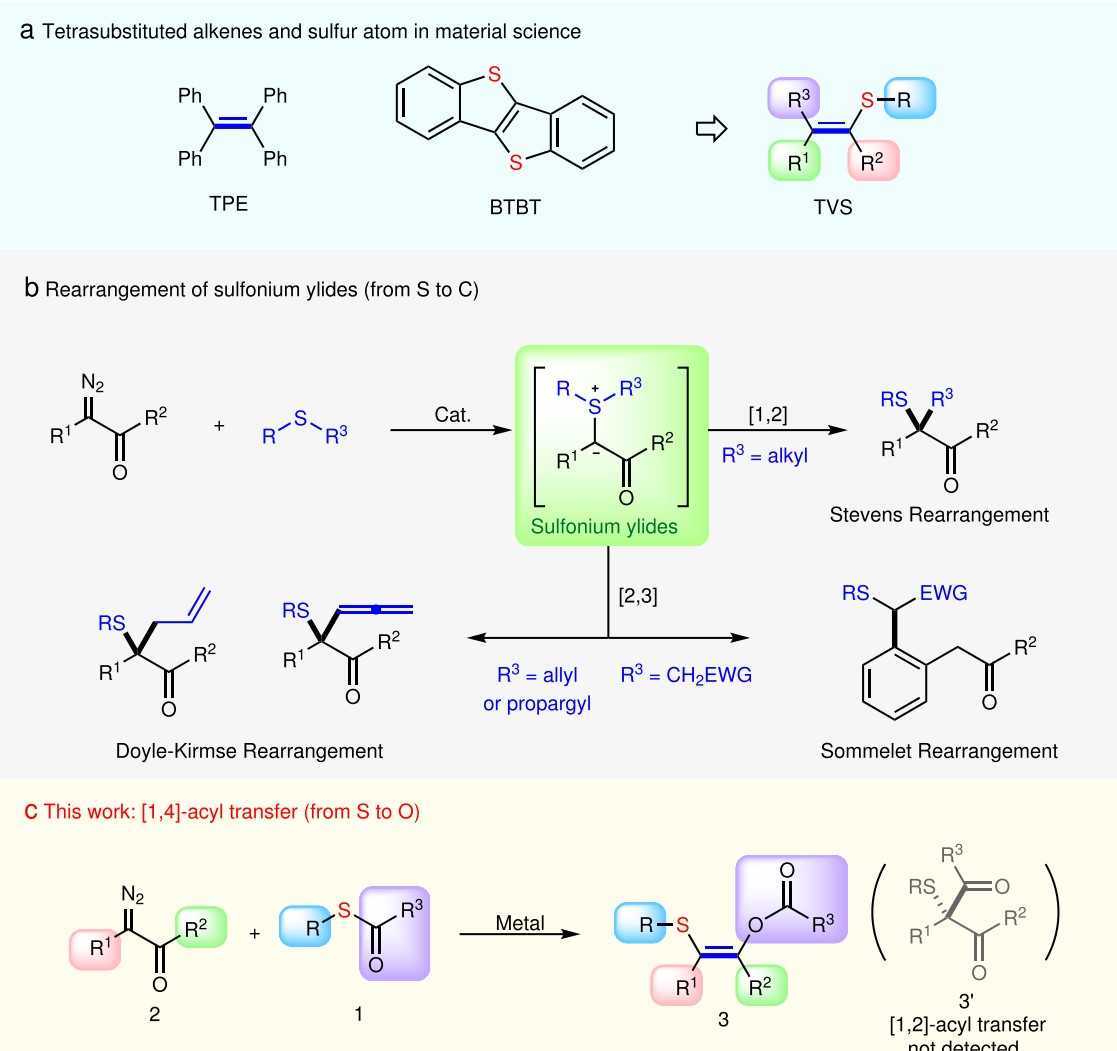

**Fig. 1 The reactions of sulfur nucleophiles and diazo carbonyl compounds. a** The design of new AIEgens. **b** Three classical rearrangement reactions via sulfur ylide. **c** 1,4-acyl transfer strategy of diazoketone and thioester. TPE: 1,1,2,2-tetraphenylethylene, BTBT: benzthieno[3,2-b]benzothiophene; TVS: tetrasubstituted vinyl sulfides.

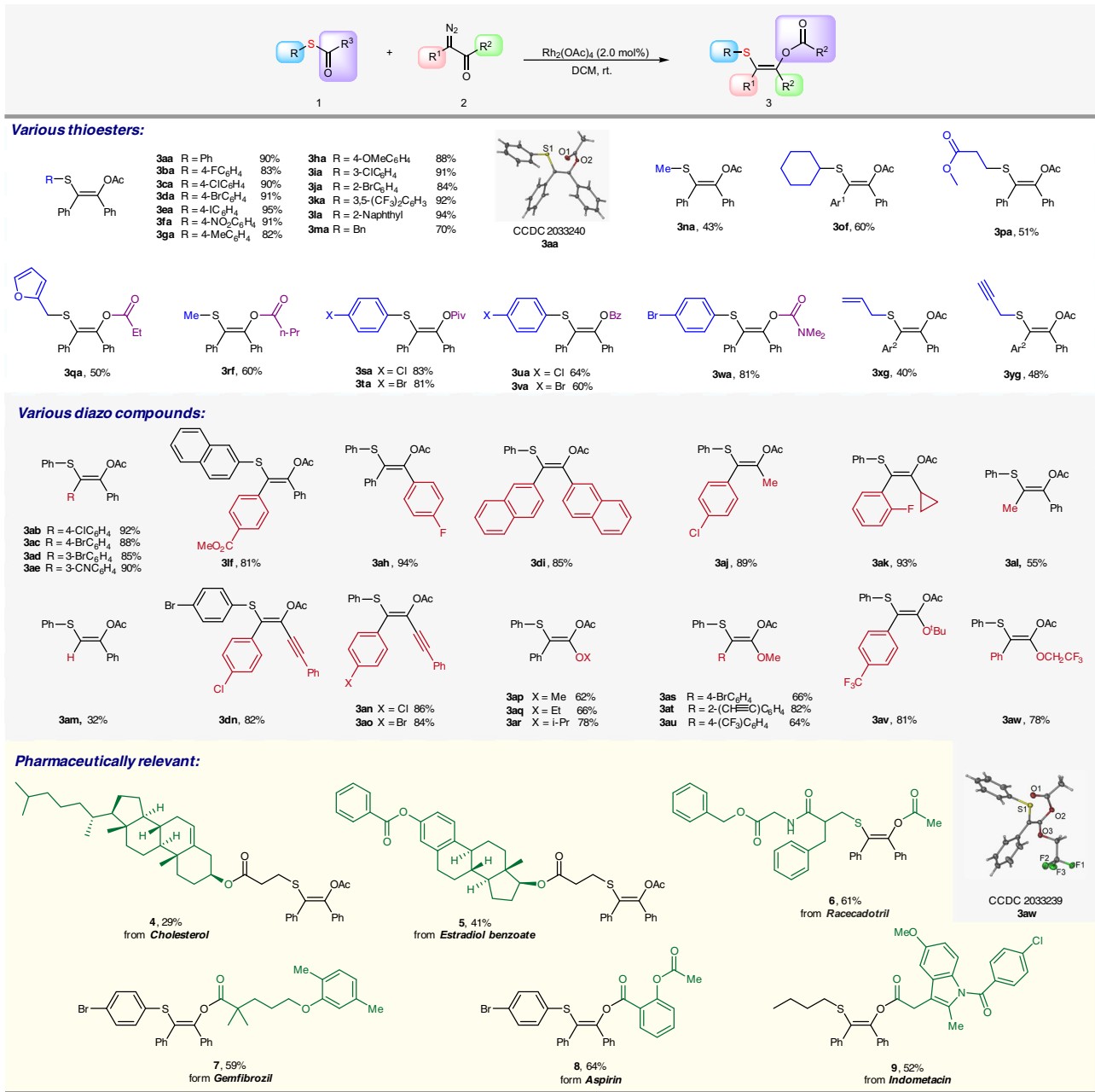

**Fig. 2 Substrate scope.** Reaction conditions: to the solution of **1** (0.5 mmol) and Rh₂(OAc)₄ (0.01 mmol, 2 mol%) in 1.0 mL DCM, the mixture of **2** (0.75 mmol) and DCM (1.0 mL) was added dropwise in 5 min at room temperature. Isolated yields for all TVSs. Ar¹ = *p*-CO₂MePh, Ar² = *m*-CO₂EtPh.

## Results

**Substrate scope of [1,4]-acyl transfer reaction**. Initially, the reaction of acetyl phenthiol **1a** and phenyl diazo ketone **2a** was chosen as the model reaction. After screening the condition, Rh₂(OAc)₄ was proven the best catalyst for this transformation (see Supplementary Table 1). We then tested the substrate scope of this [1,4]-acyl transfer reaction. As shown, a diverse range of *S*-aryl ethanethioate, with different substitutes on *para*-, *meta*-, and *ortho*-positions of phenyl ring, were suitable substrates for [1,4]-acetyl transfer reaction, affording the corresponding tetrasubstituted Z-olefins **3aa-3ka** in good to excellent yields with excellent chemo- and stereo-selectivity (Fig. 2). To our delight, the alkyl groups, including benzyl, cyclohexyl, ester-containing alkyl, on the sulfur atom were tolerated (Fig. 2, **3ma-3pa**). It was noteworthy that the well-known Stevens rearrangement had not

been observed in this transformation. When aryl thiol with various acyl groups with steric hindrance and carbamoyl group on sulfur atom were tested in this transformation, the same [1,4]-acyl and carbamoyl transfer took place, affording the corresponding Z-olefins in good efficiency (Fig. 2, **3qa-3wa**). Interestingly, attempts using the ethanethioate with allyl and propargyl on S atom exclusively delivered the acyl transfer products without any [2,3]-rearrangement products being detected (Fig. 2, **3xg-3yg**). Next, we began to study the substrate scope of various diazo carbonyl compounds **2**. Diazo ketones **2b-2o** containing various substituents, including aryl, alkyl, alkynyl, and hydrogen were compatible, delivering the expected products **3ab-3ao** in good yields. Notably, diazoesters were compatible and delivered the alkene products in good yields with single Z-selectivity (Fig. 2, **3ap-3aw**). Furthermore, this protocol was also applicable to

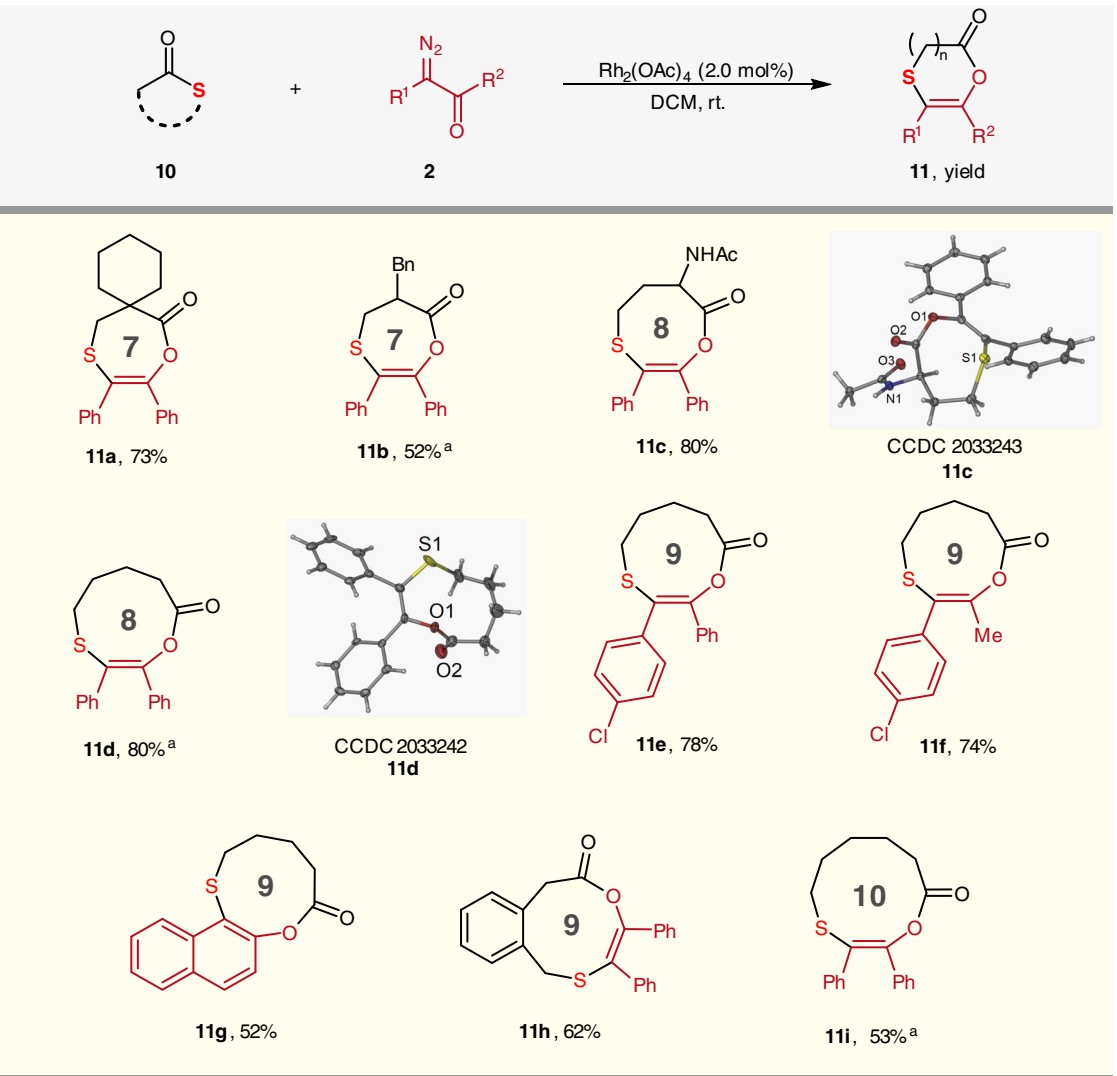

**Fig. 3 Ring expansion reaction of thiolactones 10 with diazoketones 2.** Reaction conditions: to the solution of **10** (0.5 mmol) and Rh₂(OAc)₄ (0.01 mmol, 2 mol%) in 1.0 mL DCM, the mixture of **2** (0.75 mmol) and DCM (1.0 mL) was added dropwise in 5 min at room temperature. Isolated yields for all TVSs. ᵃThe reaction scale is 0.4 mmol.

pharmaceutically relevant molecules due to the mild reaction conditions and convenient operation. To our delight, Cholesterol, Estradiol, Racecadotril, Gemfibrozil, Aspirin, and Indometacin derivatives gave the [1,4]-acyl migration products **4–9** in moderate to good yields. The structures of **3aa** and **3aw** were further confirmed by single-crystal X-ray crystallography analysis.

Then, we envisaged that the analogous [1,4]-acyl migration reaction should be applicable to the cyclic thioester substrate, which would undergo the ring expanding reaction by three atoms and afford medium-sized lactones containing an alkenyl sulfide moiety. When the spiro four-membered ring thiolactone **10a** reacted with diazoketones **2a** under the standard condition, the seven-membered lactone **11a** was generated in 73% isolated yield via the sequential ylide formation/[1,4]-acyl migration as we expected. Thus, various thiolactones **10** equipped with the rings from four-member to seven-member were tested (Fig. 3). All the reactions of **10** with diazoketone **2** were carried out smoothly, affording the corresponding lactone **11b-11i** with seven-membered to ten-membered rings in moderate to good yields. It was noteworthy that the reaction of thiolactone with amide group gave the ring expansion product **11c** without the N–H

insertion product. This result indicates this tandem ylide formation/acyl migration reaction exhibited excellent chemoselectivity, which occurred preferentially over the N–H insertion reaction. The structures of **11c** and **11e** were further confirmed by single-crystal X-ray crystallography analysis (Fig. 3).

**Gram-scale reaction and the transformations of products**. As shown in Fig. 4, this [1,4]-acyl migration reaction of thioester and diazoketones was easy to scale-up. The 5 mmol scale reaction of **2a** and **1a** was carried out under 0.1 mol% Rh₂(OAc)₄, furnishing 1.44 g of the desired product **3aa** in 83% yield. Gratifyingly, the catalyst loading could be further decreased to 0.02 mol% without effect on the yield, and the TON was up to 3650. The 22.5 mmol scale reaction of **2a** and **1d** could give 6.988 g of the corresponding product **3da** in 73% yield. It should be noted that **3da** could be obtained after simple recrystallization instead of column chromatography (see Supplementary Fig. 1 for details), demonstrating the promising synthetic practicality of this reaction. To examine the synthetic value of this protocol further, several transformations of **3aa** and **3da** were also performed (Fig. 4).

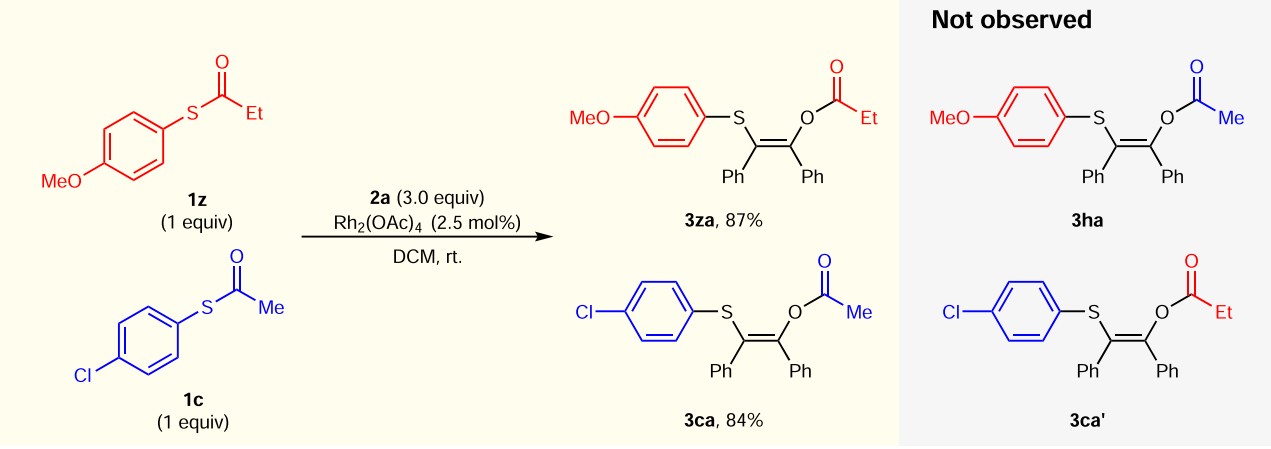

**Fig. 4 Gram-scale reaction and the transformations of products. a** *m*-CPBA (1.2 equiv), NaHCO₃ (1.0 equiv), DCM, 0 °C. **b** *m*-CPBA (5.0 equiv), NaHCO₃ (1.0 equiv), DCM, 0 °C. **c** Pd(PPh₃)₂Cl₂ (5 mol%), CuI (5 mol%), trimethylsilylacetylene (2 equiv), Et₃N, 80 °C, 3 h. **d** Pd(PPh₃)₄ (10 mol%), Na₂CO₃ (2.0 equiv), ArB(OH)₂ (1.5 equiv), toluene: H₂O = 1:1, 100 °C, 3 h.

**Fig. 5 Crossover experiment.** Reaction conditions: **1z** (0.5 mmol), **1c** (0.5 mmol), and Rh₂(OAc)₄ (0.025 mmol, 2.5 mol%) in 2.0 mL DCM, **2a** (1.5 mmol), and DCM (2.0 mL) was added dropwise in 10 min at room temperature. Isolated yields for **3za** and **3ca**.

When the **3aa** was treated with *m*-CPBA, vinyl sulfoxide (**12**) and vinyl sulfone (**13**) could be obtained in 81% and 65% yields, respectively (Fig. 4a, b). Meanwhile, the cross-coupling of **3da** with terminal alkyne and aryl boronic acid afforded the corresponding products **14–16** in moderate to excellent yields (Fig. 4c, d).

**Crossover experiment**. In order to understand the reaction mechanism, a crossover experiment was conducted (Fig. 5). As a result, the expected non-crossover products **3za** and **3ca** were obtained in excellent yields and no crossover products (**3ha** or **3ca′**) was observed in the crude ¹H NMR. This results proved this [1,4]-migration was a concerted rearrangement pathway.

**DFT calculations**. To further understand the mechanism of this reaction, density functional theory (DFT) calculations were then carried out with the Gaussian 09 software package[60–63]. The calculation details were provided in the Supplementary Information. The reaction of **2a** and **1a** was selected as the model reaction. As illustrated in Fig. 6, the reaction path included the formation of rhodium carbene intermediate **Int-1**, then the generation of free sulfur ylide **Int-3** via a metal-bound ylide intermediate **Int-2**, and finally [1,4]-shift of acetyl group to form product **3aa**, in which the formation of rhodium carbene **A** was the determining step. The barrier was 23.5 kcal/mol, which meant this reaction can proceed smoothly at room temperature. This was in good line with the experiment. The [1,4]-shift of acetyl process is almost barrierless, with a barrier of only 1.1 kcal/mol.

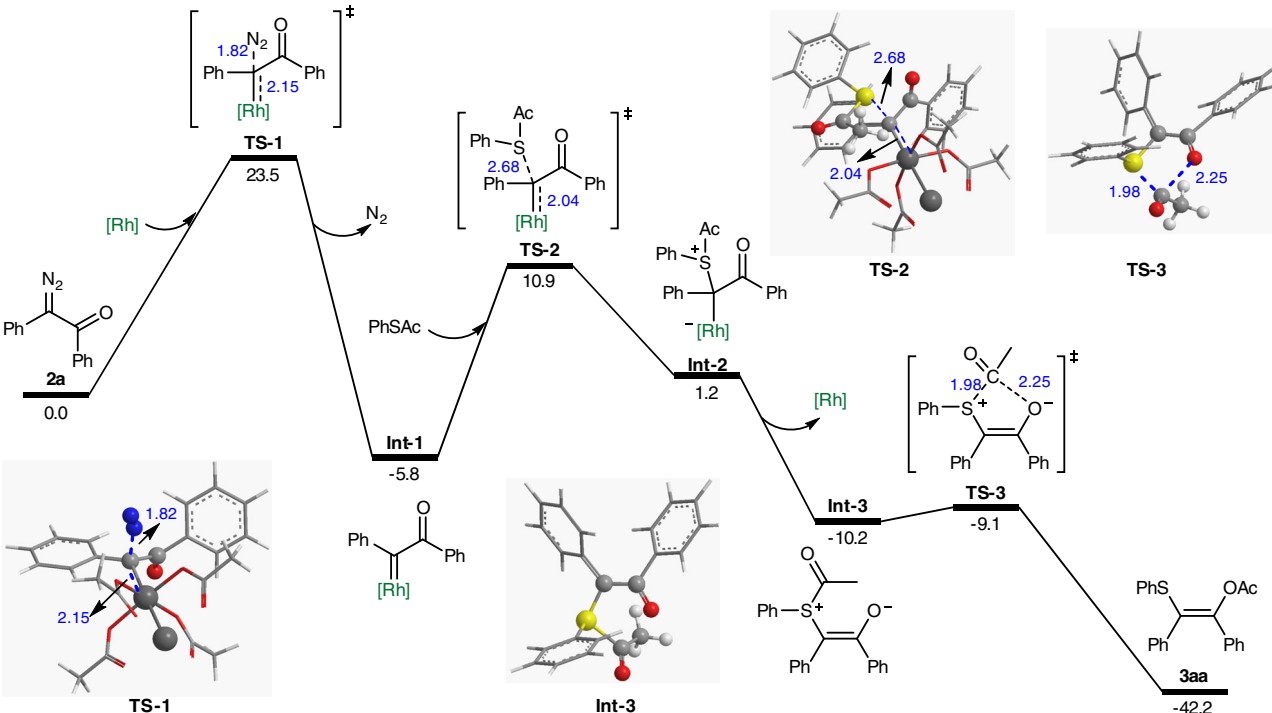

**Fig. 6 DFT calculations.** Computational exploration for the reaction of **2a** and **1a**. [Rh] = Rh$_2$(OAc)$_4$.

The rhodium carbene intermediate to coordinate with the carbonyl oxygen of the thioester **Int-2′** was located, while the relative free energy is 3.3 kcal/mol higher than that with sulfur. Furthermore, we also located a more stable rhodium-stabilized acylsulfur ylide **Int-3′** by −6.5 kcal/mol, compared to **Int-2** (see Supplementary Fig. 2 for more details). While the corresponding 1,4-acyl migration process is less favored than the free ylide, the free energy difference between two TSs (**TS-3** and **TS-3′**) is up to 6 kcal/mol. So the free ylide should be the intermediate in this reaction.

This barrier was lower than that of [2,3]-rearrangement of allyl and propargyl, which could explain the excellent chemoselectivity of this reaction (see Supplementary Figs. 3 and 4 for more details). Furthermore, this calculation demonstrated that this [1,4]-acyl migration presumably proceeds via a free ylide intermediate, which was consistent with the known [1,4]-acyl migration from O to C[64].

**Photophysical properties of TVSs.** Having established this protocol to conveniently construct the library of tetrasubstituted vinyl sulfides (TVSs), we then tested the photophysical properties of these molecules. Classic AIE was observed by dissolving **3ca** in the mixture solution of 1,4-dixoane and water with different fractions of water. Compound **3ca** showed almost no fluorescence in 1,4-dixoane. With the gradual addition of water into the solution of **3ca** in 1,4-dixoane, the emission intensity increased slightly when water fraction was less than 70%. However, when water fraction exceeded 70%, the emission intensity of **3ca** increased significantly since the restriction of intramolecular rotation (Fig. 7a, b, d). All the TVSs with *cis*-1,2-diaryl substituents exhibit this phenomenon, and the fluorescence emission spectra of representative vinyl sulfides were also measured and the emission maxima of all the AIE-gens altered from 418 to 442 nm (Fig. 7c, and see Supplementary Information for details). The results indicated the substituents on diazo compounds **2** were vital for the AIE-activity of the producing vinyl sulfides. The TVSs with two aryls or one aryl and one alkynyl exhibited AIE-activity, while the olefins containing H, alkyl, and alkoxyl did not have AIE effect. Moreover, some AIEgens, such as **3aa**, **3ca**, **3da**, **3sa**, **3dn**, and **15**, exhibted interesting luminescence in solid state (Fig. 7c, and see Supplementary Table 7 for more details). Thus, these compounds might be useful for OLEDs and living animal imaging via structural modification.

**Discussion**

In summary, we have developed a rhodium-catalyzed rearrangement reaction of thioesters and diazoketones via sulfur ylide, which provides a modular strategy to construct a library of tetrasubstituted vinyl sulfides (TVSs) in good to excellent yields. This protocol features broad substrate scope (63 examples), mild condition, low catalytic loading, convenient operation, and easy scale-up. The TON of this reaction is up to 3650 and the product can be isolated without column, indicating that this reaction might be promising tools in industry. The mechanistic studies combining experiments and DFT calculation exhibit this [1,4]-acyl shift from S to O is intramolecular pathway and has lower energy bar than known [2,3]-rearrangement via a free ylide intermediate. Furthermore, the spectroscopic properties of these molecules were studied in solution as well as solid state, and TVS is proved to be a new type of AIEgens. Some of them might be useful for OLEDs and living animal imaging via structural modification. This work would inspire the development of new sulfur-containing AIEgens and broaden the application of catalytic reactions in synthesis of materials.

**Methods**

**General procedure for TVSs.** Thioester **1** or **10** (0.5 mmol, 1.0 equiv), and Rh$_2$(OAc)$_4$ (0.01 mmol, 2.0 mol%) were introduced into a dried glass tube under N$_2$ protection, and add 1 mL dry DCM as solvent, then the diazoketone **2** (0.75 mmol, 1.5 equiv) was dissolved in 1 ml of DCM and add dropwise in 5 min at room temperature. After the addition, continue to react for 1 min consumed diazo completely determined by TLC analysis. The mixture was purified by column chromatography on silica gel using PE/EtOAc as the eluent and concentrated to obtain the product **3** or **11**. All new compounds were fully characterized (see the Supplementary Information).

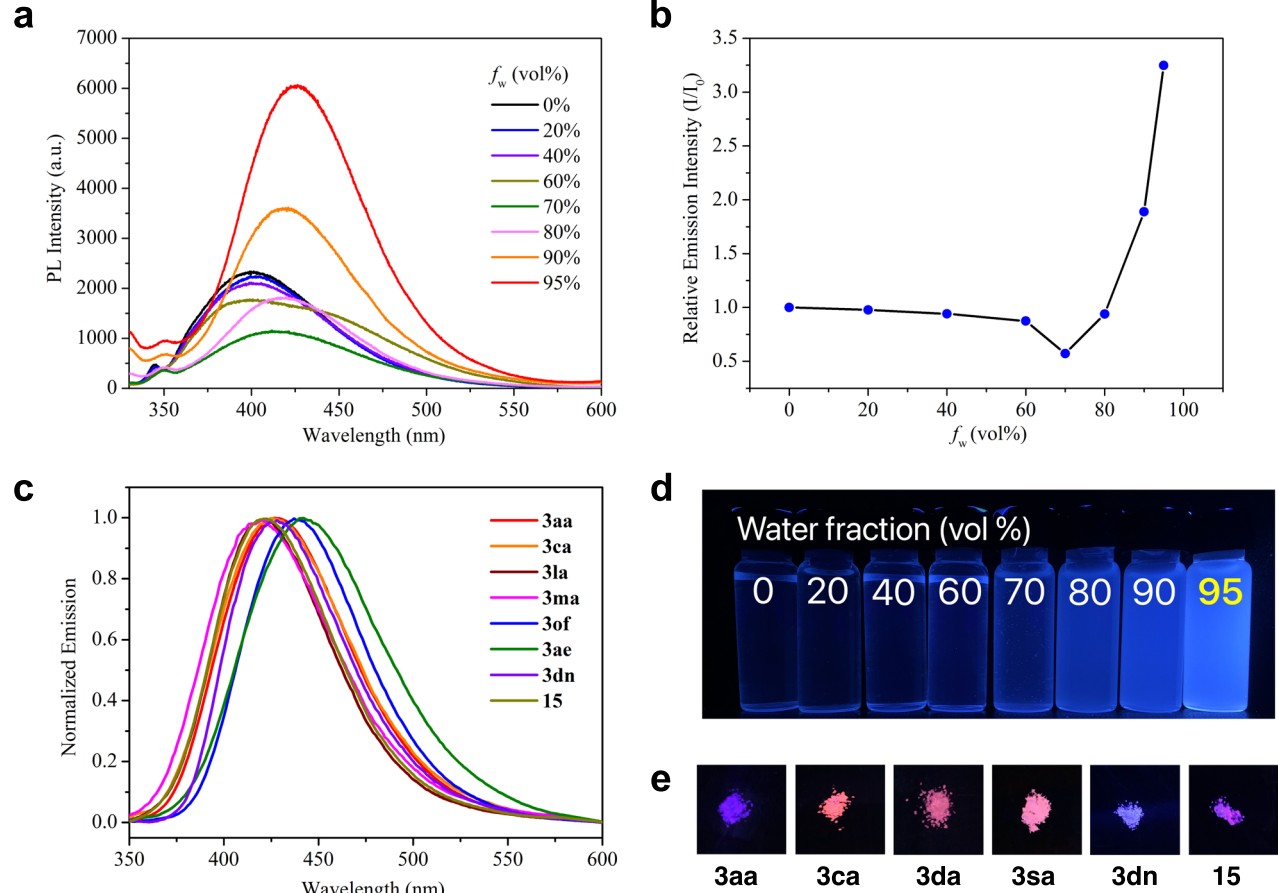

**Fig. 7 Photophysical property test. a** Emission spectra of **3ca** obtained at different water fractions of the 1,4-dixoane/water solvent mixtures. **b** Relative emission intensity of compound **3ca** a in 1,4-dioxane/H$_2$O mixture with increasing water fractions ($f_w$) to 95% ($c = 150 \, \mu M$, $\lambda_{ex} = 312 \, nm$, $\lambda_{em} = 426 \, nm$). **c** Normalized fluorescence emission spectra in 1,4-dixoane/water mixtures ($c = 150 \, \mu M$, 95% water) of selected vinyl sulfide. **d 3ca** ($c = 150 \, \mu M$) in 1,4-dixoane/water mixtures with different volume fractions of water and **e** photographs of solid compounds; all photographs taken upon excitation at 365 nm using a UV lamp at 298k.

## Data availability

All other data in support of the findings of this study are available within the Article and its Supplementary Information or from the corresponding author upon request. X-ray crystallographic data for compound **3aa** (CCDC 2033240), **3aw** (CCDC 2033239), **11c** (CCDC 2033243), and **11e** (CCDC 2033242) are freely available from the Cambridge Crystallographic Data Center. Copies of the data can be obtained free of charge from the Cambridge Crystallographic Data Center via https://www.ccdc.cam.ac.uk/data_request/cif.

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

## Acknowledgements
We are grateful to the National Natural Science Foundation of China (Nos. 21971066, 21772042) and the Science and Technology Commission of Shanghai Municipality (18JC1412300) for financial support. We greatly appreciate Dr. Qiuhua Zhao at East China normal university for her kind help with NMR test, Prof. Xiaoli Zhao at East China normal university for her kind help with X-ray single-crystal structural analyses.

## Author contributions
L.L. conceived the idea; X.-S.L. performed the most experiments; Z.T. assisted in some experiments; Z.T. and M.L. helped in synthesis of substrates **1** and **2**. X.-S.L. collected and analyzed the data; Z.L. carried out DFT calculations; L.X. guided the test of photophysical properties; L.L. guided this project and wrote the manuscript.

## Competing interests
The authors declare no competing interests.
