## [Peer Review File · Nature Communications]

REVIEWER COMMENTS

Reviewer #1 (Remarks to the Author):

The authors report an interesting and efficient chemo- and stereo-selective Rh(II)-catalyzed [1,4]-acyl rearrangement of

α -diazo carbonyl compounds and thioesters to give tetrasubstituted vinyl sulfides in overall good yield with a broad substituent scope. The process is unprecedented and can be considered as novel. Also the modular character enables providing a large library of compounds with a specific substitution pattern, where the stereoselectivity can be excellently controlled. The finding several of the compounds are AIE chromophores is somehow not too surprising, yet, some basic characterization have been provided. However, as often in purely synthesis motivated studies the more difficult findings are only vaguely explored, e.g. the occurrence of fluorescence and phosphorescence. These findings are still far from being close to rationalization. To my opinion certain solid state findings should first be scrutinized and elucidated closely before reporting them. Also the discussion part is far from being a discussion, rather a summary.

Some statements in the introduction, e.g. "Sulfur-containing motifs frequently play a vital role in organic photoelectric materials because S atom possesses higher resonance energy than that of the other heteroatoms." are bewildering and to some extent unclear. Why does sulfur possess a high resonance energy? Isn't it sulfurs polarizability and hyperpolarizability which makes it so peculiar for the design of materials.

The conclusion of the introduction with AIE might seem logical from the perspective of the phenomenological findings in this work, but they are far away from a rational scientific design. I think the synthetic, methodological part, particularly with the late stage functionalization of drugs and as a preview on one conclusiv AIE system would make this communication more sound.

In conclusion, I recommend acceptance of the purely synthetic and methodological part, omitting the inconclusive photophysics at this stage. Therefore, acceptance can only be recommended after major revisions, i.e. omitting large parts of the unexplicable photophysics. It would warrant a follow-up study with deep reaching experiments clarifying the occurrence of complex emission characteristics.

Reviewer #2 (Remarks to the Author):

In this manuscript, Liu and coworkers reported Rh(II)-catalyzed [1,4]-acyl rearrangements of α -diazo carbonyl compounds and thioesters for the synthesis of 2- tetrasubstituted vinyl sulfides. Some of these products show AIE properties, which may find applications as functional materials. The highlight of this work is the usage of [1,4]-acyl rearrangement of sulfonium ylide, which led to the cleavage of C-S bond of thioester and the formation of vinyl sulfide and vinyl ether. Moreover, this reaction works well for both noncyclic and cyclic thioester substrates with large scope of different functional groups. Notably, such transformation of cyclic thioesters can be viewed as a ring expansion reaction, enabling the access to a range of medium-or large ring lactones. The manuscript is recommended for publication in Nat. Commun. after careful revisions.

1. A similar work on Rh(II)-catalyzed [1,4]-acyl rearrangements of 3-diazoindol-2-imines with thioesters has been reported by Wang et al. (Org. Biomol. Chem., 2018, 16, 439). Please spell out the difference between this and previous papers, especially the novelty of the present work.

2. Page 3, the authors stated, "The formation of C(sp²)-S bond via the sulfonium ylides rearrangement of the reaction of sulfur-containing compounds with α -diazocarbonyl compounds is still unknown." However, in Wang's work (Org. Biomol. Chem., 2018, 16, 439), the C(sp²)-S coupling has been well investigated. Please rewrite the sentence and cite this work.

3. Many places in the whole paper, the "yield" should be corrected into "yields", like "in 81% and 65% yield", "delivering the expected products 3ab-3ao in good yield", "in moderate to excellent yield".....

4. In Page 5, the use of "It must be noted that ..." is too strong, and "it should be ..." will be better.

5. The same sentence, I am just wondering that how does "...the simple recrystallization instead of column chromatography" can conclude that..."demonstrating the promising practicality of this reaction in industry." Please delete the "in industry" and modify the sentence properly.

6. In Page 7, almost all the products synthesized shows their emission around 400-640 nm, which cannot conclude that "Considering that their near infrared emission and....". Generally, the wavelength

of near-infrared region is >780 nm.

7. It is suggested to provide a plot of relative emission intensity (I/I_0) versus the composition of the dioxane/water mixture of 3ca, which can show the AIE effect more clearly.

Reviewer #3 (Remarks to the Author):

The authors have taken a well-known transformation (Stevens Rearrangement) to a new outcome that is a 1,4-acyl transfer and in doing so is able to prepare a new type of AIEgens that may be of practical use. On reflection, the pathway to this outcome is predictable, but to my knowledge this is the first report of such a rearrangement. One might have expected that the rhodium carbene intermediate to coordinate with the carbonyl oxygen of the thioester, rather than with sulfur (it would be of value if the authors could provide the energy for this step). Once the sulfur ylide is formed, if the rhodium acetate has departed to form the free ylide, one would have expected to observe the 1,4-acyl transfer, but there is evidence in ylide reactions that the rhodium-stabilized ylide rearranges with rhodium migrating to the carbonyl oxygen of the original diazoester. In the reaction step involving formation of the rhodium-stabilized acylsulfur ylide, the authors show initial departure of rhodium acetate then 1,4-acyl migration. Could this duality be synchronous? I believe that this is an important consideration because there is increasing evidence that free ylides are not intermediates in rhodium(II) catalyzed reactions.

The scope of this transformation is extensive.

Lu Liu
School of Chemistry and Molecular Engineering,
East China Normal University,
500 Dongchuan Road, Shanghai 200241, China
Email: lliu@chem.ecnu.edu.cn
Tel: (+86)-21-54341205

Dear Referees:

Thank you very much for your kind letter. We received the comments back for the manuscript entitled “Modular and Stereoselective Synthesis of Tetrasubstituted Vinyl Sulfides Leading to a Library of AIEgens via Carbene Transfer Reactions”. We would like to express our thanks to all referees for your careful reviewing, suggestions, which helps us improve the quality of this work. We have already prepared the revised manuscript. In the following, a point by point response to the reviewer comments is given. All of our changes are in yellow in the revised manuscript.

REVIEWER COMMENTS

Reviewer #1 (Remarks to the Author):

The authors report an interesting and efficient chemo- and stereo-selective Rh(II)-catalyzed [1,4]-acyl rearrangement of α -diazo carbonyl compounds and thioesters to give tetrasubstituted vinyl sulfides in overall good yield with a broad substituent scope. The process is unprecedented and can be considered as novel. Also the modular character enables providing a large library of compounds with a specific substitution pattern, where the stereoselectivity can be excellently controlled. The finding several of the compounds are AIE chromophores is somehow not too surprising, yet, some basic characterization have been provided. However, as often in purely synthesis motivated studies the more difficult findings are only vaguely explored, e.g. the occurrence of fluorescence and phosphorescence. These findings are still far from being close to rationalization. To my opinion certain solid state findings should first be scrutinized and elucidated closely before reporting them. Also the discussion part is far from being a discussion, rather a summary.

Reply: We thank reviewer 1 for her/his kind recommendation. In this manuscript, we principally present the synthetic methodology to introduce the stereoselective synthesis

of the tetrasubstituted vinyl sulfides, which might be useful for the synthesis of a series of functional molecules. And we also present the interesting luminescence of the tetrasubstituted vinyl sulfides, which might be attractive to the scientists in the field of material. We transferred the solid state findings from manuscript to Supplementary Information.

Some statements in the introduction, e.g. "Sulfur-containing motifs frequently play a vital role in organic photoelectric materials because S atom possesses higher resonance energy than that of the other heteroatoms." are bewildering and to some extent unclear. Why does sulfur possess a high resonance energy? Isn't it sulfurs polarizability and hyperpolarizability which makes it so peculiar for the design of materials.

Reply: We thank reviewer 1 to point this out. We got the wrong information from literature (*Chem. Commun.*, **2017**, *53*, 2918). We totally agreed the reviewer's comments and revised this sentence.

The conclusion of the introduction with AIE might seem logical from the perspective of the phenomenological findings in this work, but they are far away from a rational scientific design. I think the synthetic, methodological part, particularly with the late stage functionalization of drugs and as a preview on one conclusiv AIE system would make this communication moresound.

Reply: We thank reviewer 1 for her/his kind recommendation. We rewrote the introduction part. From the literature, we knew trisubstituted vinyl sulfides AIE characteristics. We thought it was still reasonable to design the tetrasubstituted vinyl sulfides AIEgens from known trisubstituted vinyl sulfides AIEgens.

In conclusion, I recommend acceptance of the purely synthetic and methodological part, omitting the inconclusive photophysics at this stage. Therefore, acceptance can only be recommended after major revisions, i.e. omitting large parts of the unexplicable photophysics. It would warrant a follow-up study with deep reaching experiments clarifying the occurrence of complex emission characteristics.

Reply: We thank reviewer 1 for her/his kind recommendation. We omitted large parts of the unexplicable photophysics in revised manuscript.

Reviewer #2 (Remarks to the Author):

In this manuscript, Liu and coworkers reported Rh(II)-catalyzed [1,4]-acyl rearrangements of α -diazo carbonyl compounds and thioesters for the synthesis of 2-tetrasubstituted vinyl sulfides. Some of these products show AIE properties, which may find applications as functional materials. The highlight of this work is the usage of [1,4]-acyl rearrangement of sulfonium ylide, which led to the cleavage of C-S bond of thioester and the formation of vinyl sulfide and vinyl ether. Moreover, this reaction works well for both noncyclic and cyclic thioester substrates with large scope of different functional groups. Notably, such transformation of cyclic thioesters can be viewed as a ring expansion reaction, enabling the access to a range of medium-or large ring lactones. The manuscript is recommended for publication in Nat. Commun. after careful revisions.

1. A similar work on Rh(II)-catalyzed [1,4]-acyl rearrangements of 3-diazoindol-2-imines with thioesters has been reported by Wang et al. (*Org. Biomol. Chem.*, 2018, 16, 439). Please spell out the difference between this and previous papers, especially the novelty of the present work.

Reply: We thank reviewer 2 to point this out. In fact, these two works are totally different. In Wang's work, they used a special diazo compounds based on indoles. As shown below, the formation of the product via the aromatization of indol-2-imines to indole, which provided huge driving force. And the products were limited to indolyl aryl sulfides.

In this manuscript, we developed a novel and general acyl shift reaction from S to O by using the most commonly used diazo carbonyl compounds, which delivering various S-containing tetrasubstituted olefins in excellent stereoselectivity. This protocol provided an efficient tools not only for the synthesis vinyl sulfides but also for the synthesis of tetrasubstituted olefins. In addition, the products have AIE properties, which might have the potential to be used in OLEDs and living animal imaging via structural modification. We have added Wang's work in the revised manuscript.

2. Page 3, the authors stated, “The formation of C(sp²)-S bond via the sulfonium ylides rearrangement of the reaction of sulfur-containing compounds with α -diazocarbonyl compounds is still unknown.” However, in Wang’s work (*Org. Biomol. Chem.*, 2018, 16, 439), the C(sp²)-S coupling has been well investigated. Please rewrite the sentence and cite this work.

Reply: Thanks for you pointing this out. We have rewritten the sentence and cited this work.

3. Many places in the whole paper, the “yield” should be corrected into “yields”, like “in 81% and 65% yield”, “delivering the expected products **3ab–3ao** in good yield”, “in moderate to excellent yield”.

Reply: We thank reviewer 2 for her/his kind recommendation. We have switched the word in revised manuscript.

4. In Page 5, the use of “It must be noted that ...” is too strong, and “it should be ...” will be better.

Reply: We thank reviewer 2 for her/his kind recommendation. We have changed this word in revised manuscript.

5. The same sentence, I am just wondering that how does “...the simple recrystallization instead of column chromatography” can conclude that...“demonstrating the promising practicality of this reaction in industry.” Please delete the “in industry” and modify the sentence properly.

Reply: We thank reviewer 2 for her/his kind recommendation. We have corrected this sentence in revised manuscript.

6. In Page 7, almost all the products synthesized shows their emission around 400-640 nm, which cannot conclude that “Considering that their near infrared emission and...”. Generally, the wavelength of near-infrared region is >780 nm.

Reply: We thank reviewer 2 bring this to our attention. We have corrected this sentence in revised manuscript.

7. It is suggested to provide a plot of relative emission intensity (I/I_0) versus the composition of the dioxane/water mixture of **3ca**, which can show the AIE effect more clearly.

Reply: We want to thank reviewer 2 for this kind suggestion. We have added the plot of relative emission intensity (I/I_0) as Fig. 8b in revised manuscript.

Reviewer #3 (Remarks to the Author):

The authors have taken a well-known transformation (Stevens Rearrangement) to a new outcome that is a 1,4-acyl transfer and in doing so is able to prepare a new type of AIEgens that may be of practical use. On reflection, the pathway to this outcome is predictable, but to my knowledge this is the first report of such a rearrangement. One might have expected that the rhodium carbene intermediate to coordinate with the carbonyl oxygen of the thioester, rather than with sulfur (it would be of value if the authors could provide the energy for this step). Once the sulfur ylide is formed, if the rhodium acetate has departed to form the free ylide, one would have expected to observe the 1,4-acyl transfer, but there is evidence in ylide reactions that the rhodium-stabilized ylide rearranges with rhodium migrating to the carbonyl oxygen of the original diazoester. In the reaction step involving formation of the rhodium-stabilized acylsulfur ylide, the authors show initial departure of rhodium acetate then 1,4-acyl migration. Could this duality be synchronous? I believe that this is an important consideration because there is increasing evidence that free ylides are not intermediates in rhodium(II) catalyzed reactions. The scope of this transformation is extensive.

Reply: Thank you for your advice, the rhodium carbene intermediate to coordinate with the carbonyl oxygen of the thioester (**Int-2'**) was located. Though the electronic effect for carbonyl oxygen coordination may be better than sulfur coordination, the relative free energy is 3.3 kcal/mol higher than that with sulfur, probably from some weak interaction and penalty of entropy. And as you advised, we have located a more stable rhodium-stabilized acyl sulfur ylide **Int-3'** by -6.5 kcal/mol, compared to **Int-2**. While the corresponding 1,4-acyl migration process (in red) is less favored than the free ylide (in black), the free energy difference between two TSs (**TS-3** and **TS-3'**) is up to 6 kcal/mol. So the free ylide may be the intermediate in this reaction. We added this results in main text and the whole reaction pathway in Scheme S1 in Supplementary Information.

We have already revised this manuscript as requested and addressed all issues by the three reviewers and editors. And we hope this improved manuscript is up to the high standard of *Nature Communications*.

Thank you very much for your kind consideration once again.

With best regards

Sincerely yours,

Lu Liu, Prof. Dr.

REVIEWERS' COMMENTS

Reviewer #1 (Remarks to the Author):

Liu and coauthors have carefully revised the manuscript with a predominant focus on the innovative synthetic/methodological part accompanied by DFT calculations to provide a founded mechanistic rationale. In comparison to the initial submission vague speculative parts on the photophysics have been omitted and only the observed undisputable occurrence of AIE has been focused. In essence the manuscript is now well publishable in Nature Communications after some minor language issues are fixed.

Reviewer #2 (Remarks to the Author):

The authors have adequately revised their manuscript according to my previous comments and suggestions. The quality of the manuscript has been improved after the revision. I do not have further criticism of the work.

Reviewer #3 (Remarks to the Author):

Worthy of publication.

Reviewer #4 (Remarks to the Author):

The authors present a DFT study of the mechanism of the transformation that is reported. The DFT study supports the proposed mechanism. The authors also sufficiently address a prior referee comment regarding the possibility of concomitant acyl migration and rhodium displacement. Computational structures provided are too small to be fully understood (both in the main text of the paper as well as in the SI). All computational structures in the SI should be enlarged and relevant bond distances (especially bond distances of critical bond-breaking and bond-forming events in transition states) should be highlighted.

Schemes S1, S2 and S3 provide free energy profiles and relevant structures of intermediates and transition states. Since none of the structures are labelled, it is unclear to the reader which of the intermediates are being discussed. The figures should be redrawn to include labels for all intermediates and transition states.

The amount of detail provided in the supplementary information is insufficient to reproduce the computational study presented. Authors should provide additional detail on whether frequency calculations were performed to confirm the correct number of imaginary modes for all structures. In addition, the authors need to specify exactly how free energies were obtained for all structures. Given the multi-component nature of the reaction, authors should comment in particular on how entropy corrections, if any, were performed. Furthermore, all relevant energies (for intermediates and transition states) used to calculate energy barriers must be provided along with the geometries so that an interested reader can check that the results can be reproduced.

Finally, the authors should comment if possible on why they chose the specified basis sets and functionals. Since there is no relevant experimental comparison, the authors should cite any sources that perhaps describe the appropriateness of the functional and basis set used for modeling Rh-carbene reactions.

Lu Liu
School of Chemistry and Molecular Engineering,
East China Normal University,
500 Dongchuan Road, Shanghai 200241, China
Email: lliu@chem.ecnu.edu.cn
Tel: (+86)-21-54341205

Dear Editor:

Thank you very much for your kind letter. We received the comments back for the manuscript entitled “Modular and Stereoselective Synthesis of Tetrasubstituted Vinyl Sulfides Leading to a Library of AIEgens via Carbene Transfer Reactions”. We would like to express our thanks to you and all referees for the careful reviewing, suggestions, which helps us improve the quality of this work. We have already prepared the revised manuscript. In the following, a point by point response to the reviewer comments is given. All of our changes are in yellow in the revised manuscript.

REVIEWER COMMENTS

Reviewer #1 (Remarks to the Author):

Liu and coauthors have carefully revised the manuscript with a predominant focus on the innovative synthetic/methodological part accompanied by DFT calculations to provide a founded mechanistic rationale. In comparison to the initial submission vague speculative parts on the photophysics have been omitted and only the observed undisputable occurrence of AIE has been focused. In essence the manuscript is now well publishable in Nature Communications after some minor language issues are fixed.

Reply: We thank reviewer 1 for her/his kind recommendation. We have reviewed the manuscript again and revised some language issues at the same time.

Reviewer #2 (Remarks to the Author):

The authors have adequately revised their manuscript according to my previous comments and suggestions. The quality of the manuscript has been improved after the revision. I do not have further criticism of the work.

Reply: We thank reviewer 2 for her/his kind recommendation.

Reviewer #3 (Remarks to the Author):

Worthy of publication.

Reply: We thank reviewer 3 for her/his kind recommendation.

Reviewer #4 (Remarks to the Author):

The authors present a DFT study of the mechanism of the transformation that is reported. The DFT study supports the proposed mechanism. The authors also sufficiently address a prior referee comment regarding the possibility of concomitant acyl migration and rhodium displacement.

1) Computational structures provided are too small to be fully understood (both in the main text of the paper as well as in the SI). All computational structures in the SI should be enlarged and relevant bond distances (especially bond distances of critical bond-breaking and bond-forming events in transition states) should be highlighted.

Reply: We have modified our text and Supplementary Information as the referee suggested. In Fig 7, bond distances of critical bond-breaking and bond-forming events in TS2 and TS3 are highlighted. All computational structures in the Supplementary Information were redrawn and enlarged, relevant bond distances in TSs have been highlighted.

2) Schemes S1, S2 and S3 provide free energy profiles and relevant structures of intermediates and transition states. Since none of the structures are labelled, it is unclear to the reader which of the intermediates are being discussed. The figures should be redrawn to include labels for all intermediates and transition states.

Reply: As the referee advised, all intermediates and transition states have been labelled in Supplementary Figures 2-4.

The amount of detail provided in the supplementary information is insufficient to reproduce the computational study presented. Authors should provide additional detail on whether frequency calculations were performed to confirm the correct number of imaginary modes for all structures. In addition, the authors need to specify exactly how free energies were obtained for all structures. Given the multi-component nature of the reaction, authors should comment in particular on how entropy corrections, if any, were performed. Furthermore, all relevant energies (for intermediates and transition states)

used to calculate energy barriers must be provided along with the geometries so that an interested reader can check that the results can be reproduced.

Reply: Relevant computational details have been added in Supplementary Information on frequency and free energy calculation. Supplementary Table 2 were added to show relevant energies for all the intermediates and transition states. All the geometries are provided in the xyz file.

Finally, the authors should comment if possible on why they chose the specified basis sets and functionals. Since there is no relevant experimental comparison, the authors should cite any sources that perhaps describe the appropriateness of the functional and basis set used for modeling Rh-carbene reactions.

Reply: Truhlar's M062x functional have been applied in many systems, especially for those including both main group elements and transition metal. (Zhao, Y., · Truhlar, D. G. The M06 suite of density functionals for main group thermochemistry, thermochemical kinetics, noncovalent interactions, excited states, and transition elements: two new functionals and systematic testing of four M06-class functionals and 12 other functionals. *Theor. Chem. Acc.* 120, 215-241 (2008)). This reference has been included in Supplementary Information. In addition, polarization and dispersion are both considered in 6-31+G(d,p) basis set, and SDD is a relatively expensive pseudopotential. The calculation results at this level are in good line with our experiment. We also tried B3LYP functional at the same basis set, and the results frustrated us.

We have already revised this manuscript as requested and addressed all issues by the three reviewers and editors. And we hope this improved manuscript is up to the high standard of *Nature Communications*.

Thank you very much for your kind consideration once again.

With best regards

Sincerely yours,

Lu Liu, Prof. Dr.